# Pyramid Attention For Source Code Summarization

**Lei Chai** and **Ming Li***
National Key Laboratory for Novel Software Technology,
Nanjing University, Nanjing 210023, China
`{chail, lim}@lamda.nju.edu.cn`

## Abstract

This paper presents a multi-granularity method for source code summarization, which generates a concise functional description for the given code snippet. We notice that skilled programmers write and read source codes hierarchically and pay close attention to conceptual entities like statements, tokens, sub-tokens, and the mapping relations between them. The entities have specific emphasis according to their granularities, e.g., statements in coarse-granularity reveal the global logical semantics of code, and the sub-tokens in fine-granularity are more related to the textual semantics. Driven by this observation, we demonstrate that a multi-granularity formulation incorporating these conceptual entities benefit the code summarization task. Concretely, the source code is transformed into a pyramidal representation, and then a pyramid attention mechanism is applied for efficient feature aggregation among different hierarchies in it. We instantiate our multi-granularity method using the proposed pyramid attention and name it PA-former (Pyramid Attention transformer). We evaluated it on two source code summarization benchmarks where it surpasses the prior works and achieves new state-of-the-art results. Our code and data are available at `https://github.com/leichainju/pa-former`.

## 1 Introduction

Automatic source code summarization is drawing increasing attention to its promising application prospects in software development and maintenance. This task is challenging due to the complex syntax structure in programs and the arbitrariness of variable naming. Summarization methods need to take advantage of textual and grammatical information in programs to learn a meaningful representation for concise description generation. Following the early stage works [12, 30] that treated the code as a sequence of text and directly adopted the well-developed seq2seq [28] models for this task, recent leading methods [27, 25] further exploit the explicit structure (i.e., abstract syntax tree or pair-wise relations extracted from it) in programs to boost the model performance.

Current structure-based code summarization methods are either in a hybrid way [27, 1, 13, 3] or in a structure-guided way [25, 35, 9, 19]. Works in hybrid fashion encode the AST-based structure using structure-aware models like GGNN [16] or TreeLSTM [29], and then combine the structural representation with the textual representation learned by sequential models for the decoding process. The structure-guided approaches use the pair-wise relations between tokens (or sub-tokens) extracted from code structure as inductive biases to guide the learning process. While these methods achieve excellent results, they only model the source code from a single-granularity perspective, which can be further improved in a multi-granularity manner.

To understand a given code snippet, a skilled programmer hierarchically breaks it down into fine-grained entities (i.e., sub-tokens) and builds up conceptual semantics gradually from fine-grained

---

*Ming Li is the corresponding author.

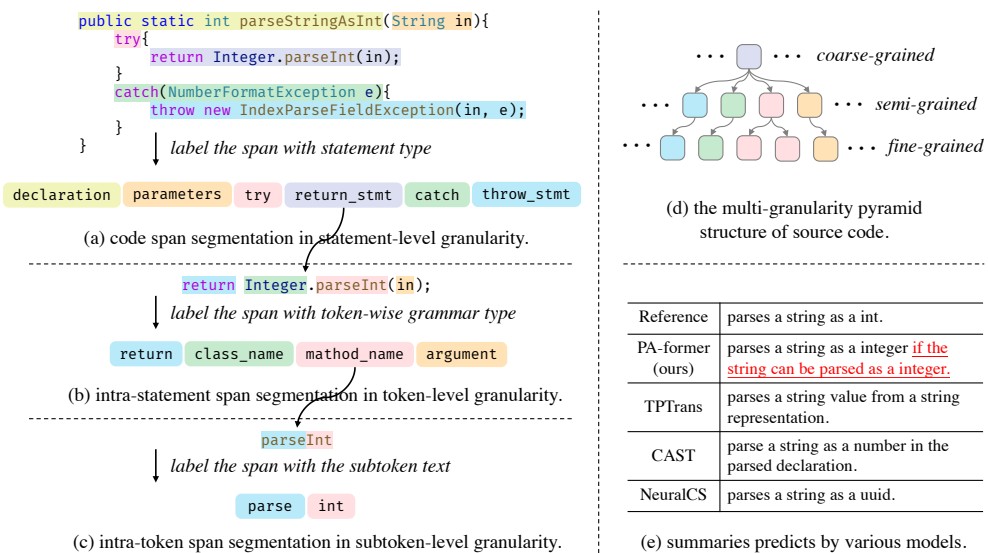

Figure 1: Example of human-like code comprehension. Skilled programmers hierarchically break the source code down to fine-grained entities (i.e., sub-tokens) and gradually build up conceptual semantics understanding for the code from fine-grained entities to coarse-grained ones.

entities to coarse-grained ones. We now use the example shown in Figure 1 to illustrate the multi-granularity process for code understanding. Concretely, we first segment the whole code into several coarse-grained spans according to punctuations and grammar, where a span corresponds to a statement conceptually (Figure 1(a)). Given the coarse-grained sequence, we can roughly summarize the code as *return something if something is done*. After that, we dive into each statement and do intra-statement segmentation for it (Figure 1(b)). Now we can detail the first *something* as *return something produced by a method invocation*. To further clarify *which method is invoked* and *what is produced*, we go deep into each token and further segment the token into sub-tokens. Here, we can figure out that *an int parsing method is invoked* and *an int is produced* (Figure 1(c)). We finally roll back the details from fine-grained entities to coarse-grained ones and obtain the overall understanding of the given code.

Driven by the above analysis, we propose our multi-granularity formulation for source code summarization by mimicking human behavior. We first construct a pyramidal input, which is divided into fine-grained layer, semi-grained layer and coarse-grained layer from bottom to top (Figure 1(d)). And a novel *pyramid attention* mechanism is proposed for efficient feature aggregation among these layers and produces a pyramidal representation with rich semantics. Because both coarse-grained logical patterns and fine-grained textual features are considered simultaneously, our model can predict a more comprehensive summary. Taking the code illustrated in Figure 1(e) as an example, compared with current leading methods, our model notices the *try ... catch ...* structure in this code snippet and uses the additional *if the string can be parsed as an int.* to tell the captured exception handling pattern.

Without bells and whistles, vanilla Transformer [31] equipped with the proposed **P**yramid **A**ttention (PA-former) achieves about 5% performance improvement (evaluated by BLEU [24]) on a Java dataset. PA-former is evaluated on two source code summarization benchmarks where it surpasses the prior works and reaches new state-of-the-art results. And ablation studies are conducted to show the efficiency of the proposed method.

## 2   Related Work

**Source code representation.**   Learning-based source code comprehension is drawing increasing attention as its promising application prospects in the field of software development and software maintenance. A meaningful representation can be used for various downstream tasks, such as code clone detection [34, 33], code classification [23, 37, 3], code summarization [10, 30, 3], code search

[7], etc. Early learning-based works treated source code as a sequence of text ignoring the structure features. Mainstream base models like CNN [2], Attention-based LSTM [12] and Transformer [30] were all used directly to capture the textual patterns in programs.

Many follow-ups leveraged both textual and structural information to learn a more meaningful code representation. Tree-based models like TBCNN [23], Code-GRU [17], ASTNN [37] were proposed to represent the source code by considering the tree structure. Some works [1] extended ASTs into graphs by introducing control flows and data flows, then a GGNN was applied to represent the code graphs. The set of paths [3, 4] extracted from AST was also used as a representation of the corresponding code. Some works [11, 8, 10] flatten the ASTs into a sequence, and then the off-the-shelf sequential models are applied to learn the textual and syntax features. Currently, most attention is focused on guiding the attention computation in Transformer by using the structural bias in programs [25, 9, 38].

**Source code summarization.** To generate a functional description for the given code snippet, code summarization models should capture both textual and structural information in programs. Some early works [12, 30, 32] just treated the source code as a sequence or flatten the AST using a structure-based traversal (SBT) method [10] and then performed a conventional seq2seq learning. The hybrid works represented both textual and structural features separately, then a hybrid mechanism is used to combine the learned representations for the summary generation. HDeepcom [11] applied the SBT method for structural representation and CAST [27] learning the structure via hierarchical splitting and reconstruction of abstract syntax trees. Many Transformer-based methods [27, 1, 13, 3] used structure information to guide the attention computation. TPTrans [25] encoded the pairwise path between code tokens and incorporated the path embeddings like relative position encoding. Unlike the above-mentioned methods which view the source code as single-granularity data, we propose a multi-granularity source code summarization method.

**Multi-scale feature learning.** The multi-scale feature processing is widely used in the computer vision community. FPN [21] leverages the pyramid-shaped features of ConvNet and creates a feature pyramid that has rich semantics at all scales. Recently, the multi-scale feature hierarchies were also introduced into Vision Transformer (ViT) by a variety of works like MViT [6], Swin [22] and HRFormer [36], which show strong evidence that Transformer could work well with the multi-scale features.

## 3  Approach

As shown in Figure 2, given a source code, we first construct sequences with three granularities and form them into a pyramid-shaped structure based on the mapping constraints between different granularities. After that, the proposed pyramid attention mechanism (in Figure 3) is applied layer by layer in the encoding stage to produce a feature pyramid which is used by a vanilla Transformer decoder to generate the summary. The details of the proposed pyramidal input constructor and pyramid attention mechanism are introduced in this section.

### 3.1  Pyramidal Input Constructor

The pyramidal input is constructed following the human code comprehension process described in Section 1. Entities in different granularities have their own emphases, telling specific attributes of source code from different viewpoints. For instance, the coarse-grained statement sequence reveals the global logical semantics of code, the semi-grained token sequence shows the code grammar details, while the fine-grained sub-token split from each token is attached with textual information. We introduce the details of the proposed pyramid input constructor in the following.

**Fine-grained sequence.** Each leaf node in the AST corresponds to a token (like `boolean`, `value`) in the source code. Since method names and variable names in programs are usually composed of multiple natural language words with *CamelCase* or *snake_case*, we can further split these combined tokens into sub-tokens, which reveal the textual information of source code. To construct the fine-grained layer (i.e., sub-token sequence), we first obtain the leaf nodes sequence of AST by depth-first pre-order traversal and then split each token into sub-tokens. Such a sub-token technique is widely used by source code summarization methods [30, 25, 27] and proved to be effective. Formally, let

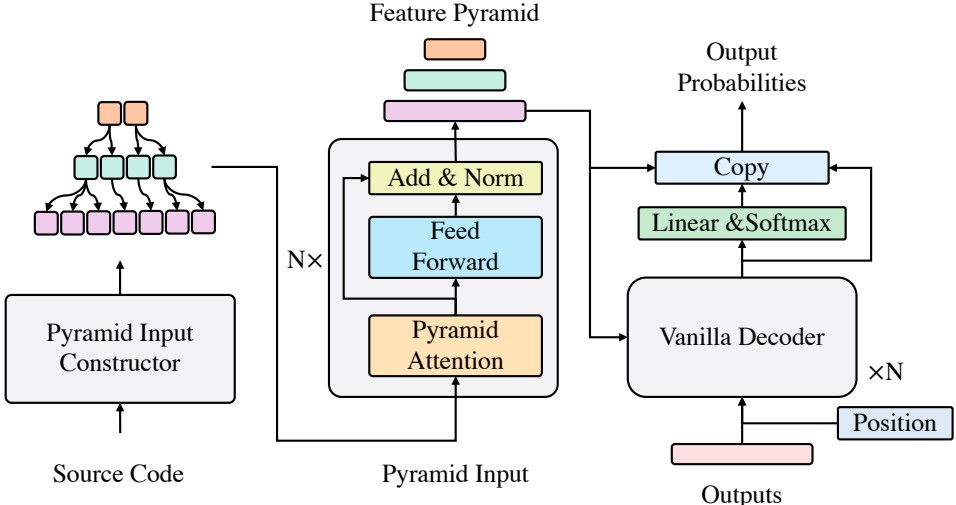

Figure 2: The proposed multi-granularity source code summarization pipeline. We first construct a pyramidal input with three granularities, and then a novel pyramid attention based encoder is applied to produce a feature pyramid which is used for decoding.

$\mathcal{T} = [t_1, t_2, \ldots, t_N]$ ($\mathcal{T}$ for text) denote the ordered sub-token sequence, that is, the fine-grained input representation situated in the bottom of the pyramidal input. For the example in Figure 1, $\mathcal{T}$ is `[public, static, int, parse, String, As, Int, (, ...]`.

**Semi-grained sequence.** The attribute of the parent node of each token (i.e. leaf) in parsed AST represents its grammatical property. We construct the semi-grained sequence by replacing the element in ordered token sequence with its grammatical property directly. Formally, let $\mathcal{G} = [g_1, g_2, \ldots, g_M]$ ($\mathcal{G}$ for grammar) denote the ordered token grammatical sequence, that is, the semi-grained input representation situated in the middle of the pyramidal input. For the example in Figure 1, $\mathcal{G}$ is `[modifier, modifier, type, identifier, (, ...]`. In addition, a one-to-many mapping relationship $\mathcal{M}_{\mathcal{G} \rightarrow \mathcal{T}}$ between token and sub-token is introduced. $\mathcal{M}_{\mathcal{G} \rightarrow \mathcal{T}}(g_i, t_j) = 1$ if the sub-token $t_j$ is split from $g_i$, which will be used to guide the information aggregation process from fine-grained hierarchy to semi-grained hierarchy.

**Coarse-grained sequence.** We use the nodes corresponding to statements to form our coarse-grained sequence, which reveals the logical information of the source code. As the statement type is limited, we directly identify the AST node as a statement-level node if the tag belongs to the pre-defined statement type set (see Appendix C for more details). Formally, let $\mathcal{L} = [l_1, l_2, \ldots, l_K]$ ($\mathcal{L}$ for logic) denote the sequence of the ordered statements, that is, the coarse-grained input representation situated in the top of the pyramidal input. For the code in Figure 1, $\mathcal{L}$ is `[declaration, parameters, try, return_statement, ...]`. Similar to the semi-grained sequence, we also construct a one-to-many mapping function $\mathcal{M}_{\mathcal{L} \rightarrow \mathcal{G}}$ between statement and token, which bridges the coarse-grained hierarchy with the semi-grained hierarchy. $\mathcal{M}_{\mathcal{L} \rightarrow \mathcal{G}}(l_i, g_j) = 1$ if the statement $l_i$ is the most recent ancestor of token $g_j$ among $\mathcal{L}$ in AST.

In summary, we construct a sequence set $\{\mathcal{T}, \mathcal{G}, \mathcal{L}\}$ in three granularities which helps the model to capture patterns from different scales and extract mapping relationships $\{\mathcal{M}_{\mathcal{L} \rightarrow \mathcal{G}}, \mathcal{M}_{\mathcal{G} \rightarrow \mathcal{T}}\}$ between adjacent granularities which introduce inductive bias to aid the model to learn accurate semantics. Thus, the sequences and the mapping relationships all together form the pyramidal input representation of the given code.

## 3.2 Pyramid Attention

The pyramidal layers $\{\mathcal{T}, \mathcal{G}, \mathcal{L}\}$ complement each other following the mapping relationships $\{\mathcal{M}_{\mathcal{L} \rightarrow \mathcal{G}}, \mathcal{M}_{\mathcal{G} \rightarrow \mathcal{T}}\}$. We propose a novel pyramid attention mechanism for efficient feature ag-

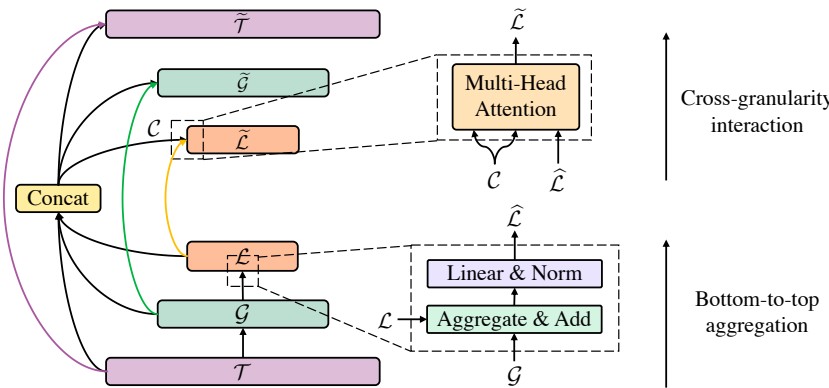

Figure 3: Pyramid attention mechanism. The pyramid attention takes a pyramid-shaped data described in Section 3.1 as input, and outputs a multi-scale representation without changing the original pyramid shape. The whole process consists of a bottom-to-top aggregation pathway and a cross-granularity information interaction process.

gregation among these layers. As illustrated in Figure 3, the pyramid attention module includes a bottom-to-top aggregation pathway and a cross-granularity information interaction process. We describe the details in the following.

**Bottom-to-top aggregation.** The bottom-to-top pathway does a fine-to-coarse information aggregation, which ensures to enrich the higher-level representation using the lower-level semantics. Given the pyramid input data: $\{\mathcal{T}, \mathcal{G}, \mathcal{L}\}$ and the mapping function $\{\mathcal{M}_{\mathcal{L} \to \mathcal{G}}, \mathcal{M}_{\mathcal{G} \to \mathcal{T}}\}$, we aggregate the representation of $\mathcal{T}$ into $\mathcal{G}$ w.r.t. $\mathcal{M}_{\mathcal{G} \to \mathcal{T}}$. The formulation of the aggregation process is given as:

$$\widehat{\boldsymbol{g}}_i = \text{LayerNorm}\left(\boldsymbol{g}_i + \boldsymbol{W}_{\text{agg}} \frac{\sum_{j=1}^{n} \mathcal{M}_{\mathcal{G} \to \mathcal{T}}(\boldsymbol{g}_i, \boldsymbol{t}_j)\boldsymbol{t}_j}{\sum_{j=1}^{n} \mathcal{M}_{\mathcal{G} \to \mathcal{T}}(\boldsymbol{g}_i, \boldsymbol{t}_j)} + \boldsymbol{b}_{\text{agg}}\right) \tag{1}$$

Where $\boldsymbol{W}_{\text{agg}} \in \mathbb{R}^{D \times D}, b_{\text{agg}} \in \mathbb{R}^D$ are learnable parameters, $\boldsymbol{g}_i, \boldsymbol{t}_i \in \mathbb{R}^D$ are the $i$-th representation vectors in $\mathcal{G}$ and $\mathcal{T}$ respectively, $D$ represents the dimension of the representation, and $\widehat{\boldsymbol{g}}_i$ is the aggregated representation of $\boldsymbol{g}_i$. The aggregation process from $\mathcal{G}$ into $\mathcal{L}$ w.r.t. $\mathcal{M}_{\mathcal{L} \to \mathcal{G}}$ is the same as above. We further conduct a aggregation directly from $\mathcal{T}$ into $\mathcal{L}$ w.r.t. $\mathcal{M}_{\mathcal{L} \to \mathcal{T}}$, which is obtained by:

$$\mathcal{M}_{\mathcal{L} \to \mathcal{T}}(\boldsymbol{l}_i, \boldsymbol{t}_j) = \vee_{k \in \{1, \dots, m\}} \left(\mathcal{M}_{\mathcal{L} \to \mathcal{G}}(\boldsymbol{l}_i, \boldsymbol{g}_k) \wedge \mathcal{M}_{\mathcal{G} \to \mathcal{T}}(\boldsymbol{g}_k, \boldsymbol{t}_j)\right) \tag{2}$$

In summary, we update $\{\mathcal{G}, \mathcal{L}\}$ into $\{\widehat{\mathcal{G}}, \widehat{\mathcal{L}}\}$ using $\mathcal{T}$ and $\{\mathcal{T}, \mathcal{G}\}$ respectively. The mapping-based aggregation is reasonable and necessary as the semantics of the element at high levels come from the combination of its descendants' semantics. The semi-grained sequence and coarse-grained sequence are initialized as the grammatical type of the tokens and statements. Figuratively speaking, they are like torsos with empty skeletons, and we need to fill them with textual information to make them complete. There is an ablation experiment demonstrating its effectiveness in Section 4.3.

**Cross-granularity interaction.** The aggregated semantics of the elements in semi/coarse-grained sequences tend to be local. To alleviate this dilemma, we further make the elements in a specific granularity to query all elements at three granularities, which guarantees that the model can capture both fine-grained and coarse-grained features.

Formally, we now have $\{\mathcal{T}, \widehat{\mathcal{G}}, \widehat{\mathcal{L}}\}$ after aggregation. Taking sequence $\widehat{\mathcal{G}} \in \mathbb{R}^{M \times D}$ of length $M$ in certain granularity as an example, $\mathcal{G}$ will query all of $\{\mathcal{T}, \widehat{\mathcal{G}}, \widehat{\mathcal{L}}\}$ in the cross-granularity interaction process. We first concatenate the pyramid representation as

$$\mathcal{C} = \text{Concat}([\mathcal{T}, \widehat{\mathcal{G}}, \widehat{\mathcal{L}}]) \in \mathbb{R}^{(K+M+N) \times D} \tag{3}$$

here $N, M, K$ are the sequence lengths of $\mathcal{T}, \widehat{\mathcal{G}}, \widehat{\mathcal{L}}$ respectively. A relative position encoded multi-head attention [26] is applied here and we use the relative position map $\mathcal{R}^{\widehat{\mathcal{G}}} \in \mathbb{Z}^{M \times |\mathcal{C}|}$ to make the

model granular-aware. The multi-head cross-attention (MHCA) process is calculated as:

$$\widetilde{\mathcal{G}} = \widehat{\mathcal{G}} + \left[\mathrm{MultiHead}(\widehat{\mathcal{G}},\mathcal{C})\right]\mathbf{W}_o \in \mathbb{R}^{M \times D} \tag{4}$$

$$\mathrm{MultiHead}(\widehat{\mathcal{G}},\mathcal{C}) = \mathrm{Concat}([\mathrm{Head}_1(\widehat{\mathcal{G}},\mathcal{C}),\dots,\mathrm{Head}_H(\widehat{\mathcal{G}},\mathcal{C})]) \in \mathbb{R}^{M \times D} \tag{5}$$

$$\mathrm{Head}_h(\widehat{\mathcal{G}},\mathcal{C}) = [\boldsymbol{z}_1, \boldsymbol{z}_2, \dots, \boldsymbol{z}_M] \in \mathbb{R}^{M \times \frac{D}{H}} \tag{6}$$

$$\boldsymbol{z}_i = \sum_{j=1}^{|\mathcal{C}|} \frac{\exp \boldsymbol{\alpha}_{ij}}{\sum_{j'=1}^{|\mathcal{C}|} \exp \boldsymbol{\alpha}_{ij'}} \left(\mathcal{C}_j \mathbf{W}_v^h + \widehat{\mathcal{R}}_{ij} \mathbf{W}_{rel\_v}^{\widehat{\mathcal{G}}}\right) \in \mathbb{R}^{\frac{D}{H}} \tag{7}$$

$$\boldsymbol{\alpha}_{ij} = \frac{\widehat{\mathcal{G}}_i \mathbf{W}_q^h}{\sqrt{D/H}} \left(\mathcal{C}_j \mathbf{W}_k^h + \widehat{\mathcal{R}}_{ij} \mathbf{W}_{rel\_k}^{\mathcal{G}}\right)^{\top} \in \mathbb{R} \tag{8}$$

$$\widehat{\mathcal{R}} = \mathrm{Embedding}_{rel}^{\widehat{\mathcal{G}}}(\mathcal{R}^{\widehat{\mathcal{G}}}) \in \mathbb{R}^{M \times |\mathcal{C}| \times \frac{D}{H}} \tag{9}$$

where $\mathbf{W}_q^h, \mathbf{W}_k^h, \mathbf{W}_v^h \in \mathbb{R}^{\frac{D}{H} \times \frac{D}{H}}, h \in [1,\dots,H]$ are projection matrices of Query, Key and Value for $h$-th head. $\mathbf{W}_o \in \mathbb{R}^{D \times D}$ is projection matrix for attention output. $H$ represents the number of heads and $\widehat{\mathcal{X}}$ represents the output of multi-head self-attention process. $\mathbf{W}_{rel\_k}^{\widehat{\mathcal{G}}}, \mathbf{W}_{rel\_k}^{\widehat{\mathcal{G}}} \in \mathbb{R}^{\frac{D}{H} \times \frac{D}{H}}$ are projection matrices of Key and Value of relative position embeddings, the $\widehat{\mathcal{G}}$ means that they are specific to $\widehat{\mathcal{G}}$.

Figure 4: a toy example of granular-aware relative position map for $\mathcal{G}$ cross-granularity interaction.

For the construction of relative position map $\mathcal{R}$, we calculate the clipped relative distance with max distance as 32 for the pair within granularity and assign the relative distance as a default value (e.g. max distance + 1 and max distance + 2) for all the pair across the specific granularity. Figure 4 shows a toy example of granular-aware relative position map with max distance as 3 for $\mathcal{G}$ cross-granularity interaction. The cross-granularity interaction for $\mathcal{T}$ and $\mathcal{L}$ is similar to $\widehat{\mathcal{G}}$. We share attention parameters (i.e. $\mathbf{W}_q^h, \mathbf{W}_k^h, \mathbf{W}_v^h$ and $\mathbf{W}_o$) across three granularities, but assign specific relative path parameters for each granularity. In summary, in this cross-granularity interaction process, the semantics of the feature pyramid are updated as:

$$\widetilde{\mathcal{L}}, \widetilde{\mathcal{G}}, \widetilde{\mathcal{T}} = \mathrm{MHCA}(\widehat{\mathcal{L}},\mathcal{C}), \mathrm{MHCA}(\widehat{\mathcal{G}},\mathcal{C}), \mathrm{MHCA}(\mathcal{T},\mathcal{C}) \tag{10}$$

### 3.3 Pyramid Attention Transformer

As illustrated in Figure 2, the overall model is similar to the vanilla Transformer. The self-attention module in the encoder is replaced with the proposed pyramid attention module for an efficient process of the multi-scale input. Notably, we only use the fine-grained sequence in the pyramidal representation for decoding, which already contains enough semantics for summary generation and reduces the complexity of the model design. Following prior works [30], we also apply a copy mechanism in our model which reduces the risk caused by out-of-vocabulary and speeds up training convergence.

Although there is overlap between the vocabularies of different granularities, the meanings of these common words are quite different when differentiated at different granularities. For instance, else may be just a natural word in the textual sequence, but a keyword for control in a grammatical sequence. Thus, we assign individual embedding layers for each granularity to avoid such confusion.

For efficient parameter usage, a parameter sharing strategy is also applied in the cross-granularity interaction process as stated in Section 3.2, We find the self-attention mechanism is strong enough

to handle this multi-role mission. Thus, our model doesn't add too many parameters compared to Transformer. What's more, such a parameter-sharing strategy also brings efficient model implementation. We only need to concatenate these three sequences and the relative position maps separately, then directly adopt a relative position encoded multi-head self-attention layer to implement the cross-granularity interaction module.

## 4 Experiments

### 4.1 Experiments Setup

**Dataset and Metrics.** To demonstrate the effectiveness of the proposed method, we conduct experiments on two widely-used and well-developed java datasets: EMSE-DeepCom[2] [11] which is collected from GitHubs Java repositories and FunCom[3] [14] which has $\sim$2 million java method-comment pairs. We filter out the examples that cannot be parsed properly or are of too large size for both datasets. And deduplication is done on EMSE-DeepCom to avoid data leakage. Table 1 shows the statistics of the datasets.

Table 1: Dataset statistics. #train and #test mean the number of examples for training and testing, #sub-token means the max sequence length when the code is tokenized as sub-token, and the rest are similar to this.

| Name | #train | #test | #sub-token | #token | #statement | #summary |
|---|---|---|---|---|---|---|
| EMSE-Deepcom | 295,967 | 12,226 | 196 | 160 | 16 | 24 |
| FunCom | 1,017,964 | 53,936 | 256 | 196 | 32 | 32 |

We use 6 metrics to evaluate the model performance on the datasets. BLEU [24], Rouge-L [20] and Meteor [5] are widely used to evaluate the quality of text similarity in the domain of natural language generation. Another 3 metrics are used to qualify the token-wise prediction performance. We use beam search with a beam size of 3 for all models in the inference phase.

**Data preprocessing.** All data preprocessing of code is based on the parsed syntax tree. We obtain the AST using the open-sourced tool Tree-sitter[4]. Please refer to 3.1 for details of the proposed pyramid data construction. We re-implement the data preprocessing pipeline for all compared baselines based on the AST parsed by Tree-sitter, please refer to Appendix C for more details.

**Baselines.** Early works viewed source code as a sequence of text and used the sequential model like LSTM or Transformer, we mainly compare with CODE-NN [12] and NeuralCodeSum [30] for this type of methods. A few works only learned the structure features for this generation task, we compare with TreeLSTM [29] here. Most works exploited the textual and structural information for this task. One group of works learned to represent these two types of information separately, then a hybrid mechanism is used to combine them in the decoding phase. We do comparisons with HDeepCom [11], ASTAttnGRU [13] and CAST [27] for the hybrid fashion. Another line of methods used structure relations as an inductive bias to guide the information interactions in the model. Here, we compare with SiT [35], GREAT [9] and TPTrans [25]. For clarity, the backbone and taxonomy of each method are listed in Table 2.

And a copy mechanism is added for all transformer-based models. We directly adopt the model implementation script if the code is publicly available, and re-implement the method according to the description in the corresponding paper if the code is not provided or not available. What's more, all baselines are integrated under a unified PyTorch-based code summarization framework developed by ourselves for fair comparisons. We provide implement details for baselines in Appendix A.3.

**Hyperparameters and training setup.** For fair comparisons, all the Transformer-based models use the default Transformer configurations with embedding dimension as 512, feedforward dimension as 2048, head number as 8, and layer number for encoder/decoder as 6 and all RNN-based

---

[2]https://github.com/xing-hu/EMSE-DeepCom
[3]http://leclair.tech/data/funcom/
[4]https://tree-sitter.github.io/tree-sitter/

Table 2: Comparisons with other code summarization methods on a middle-size dataset and a large-size dataset.

| Methods | Backbone | Hybrid | Structure | Epochs | BLEU | Rouge-L | Meteor | Precision | Recall | F1 |
|---|---|---|---|---|---|---|---|---|---|---|
| RMSE-Deepcom (middle-size) | | | | | | | | | | |
| CODE-NN [12] | LSTM | | | 24 | 28.448 | 43.506 | 17.886 | 47.266 | 45.165 | 44.774 |
| TreeLSTM [29] | LSTM | | ✓ | 24 | 28.992 | 43.985 | 18.179 | 48.486 | 45.115 | 45.287 |
| HDeepCom [11] | GRU | ✓ | ✓ | 24 | 32.179 | 49.029 | 21.528 | 54.273 | 50.584 | 50.754 |
| ASTAttnGRU [13] | GRU | ✓ | ✓ | 24 | 33.041 | 49.761 | 22.205 | 54.539 | 51.754 | 51.467 |
| SiT [35] | Transformer | | ✓ | 30 | 35.689 | 53.750 | 24.196 | 60.784 | 54.243 | 55.717 |
| GREAT [9] | Transformer | | ✓ | 30 | 36.382 | 53.606 | 24.181 | 60.008 | 54.230 | 55.460 |
| NeuralCodeSum [30] | Transformer | | | 30 | 37.133 | 54.800 | 25.051 | 61.356 | 55.413 | 56.682 |
| CAST [27] | Transformer | ✓ | ✓ | 30 | 37.195 | 54.868 | 25.069 | 61.601 | 55.368 | 56.747 |
| TPTrans [25] | Transformer | | ✓ | 30 | 37.248 | 54.996 | 25.022 | 62.022 | 55.354 | 56.884 |
| PA-former (ours) | Transformer | | | 30 | **38.848** | **56.095** | **25.895** | **62.498** | **56.642** | **57.903** |
| Funcom (large-size) | | | | | | | | | | |
| CODE-NN [12] | LSTM | | | 24 | 31.862 | 48.896 | 19.108 | 54.018 | 49.241 | 49.924 |
| TreeLSTM [29] | LSTM | | ✓ | 24 | 31.462 | 48.293 | 18.869 | 53.237 | 48.902 | 49.332 |
| HDeepCom [11] | GRU | ✓ | ✓ | 24 | 35.063 | 53.350 | 22.645 | 59.460 | 53.693 | 54.808 |
| ASTAttnGRU [13] | GRU | ✓ | ✓ | 24 | 37.001 | 55.034 | 23.753 | 61.244 | 55.315 | 56.523 |
| SiT [35] | Transformer | | ✓ | 36 | 42.121 | 59.330 | 26.819 | 65.562 | 59.427 | 60.839 |
| GREAT [9] | Transformer | | ✓ | 36 | 43.286 | 60.364 | 27.439 | 66.186 | 60.580 | 61.831 |
| NeuralCodeSum [30] | Transformer | | | 36 | 43.355 | 60.405 | 27.540 | 66.488 | 60.459 | 61.860 |
| TPTrans [25] | Transformer | | ✓ | 36 | 43.450 | 60.566 | 27.607 | 66.765 | 60.568 | 62.030 |
| CAST [27] | Transformer | ✓ | ✓ | 36 | 43.580 | 60.524 | 27.665 | 66.458 | 60.657 | 61.976 |
| PA-former (ours) | Transformer | | | 36 | **44.649** | **61.450** | **28.274** | **67.210** | **61.593** | **62.863** |

models use the hidden dimension with 512. For embeddings, we use the embedding dimension with 512 for all approaches. The size of vocabularies is limited by 50000 over code sequences and 30000 over summary sequences in EMSE-DeepCom, and the corresponding configurations are 35000 and 30000 in FunCom. All models are trained using NVIDIA Tesla A100 GPUs with a batch size of 64. We train all baselines including our models using AdamW optimizer with a multi_step learning rate scheduler, and set the initial learning rate to 0.0002 and 0.003 for Transformer-based and RNN-based models, respectively. And warmup strategy is used for stable training in the Transformer-based model. The training epochs for each method are listed in Table 2.

## 4.2 Main Results

**Comparisons with baselines.** The comparisons with baselines are listed in Table 2. Without bells and whistles, we find that our proposed PA-former substantially surpasses all baselines and achieves new state-of-the-art on both middle-size and large-size datasets. Compared with NeuralCodeSum which is a relative positional encoded Transformer, our method has a 5% improvement in BLEU on the middle-size dataset. For the current leading methods CAST and TPTrans, which both take all the nodes in ASTs and the sub-token sequence as input, our model only takes a subset of AST nodes and sub-tokens surpass both of them over 1.0+ BLEU scores on two datasets. And thanks to the efficient implementation and parameter sharing strategy, our model requires far less training time to converge, while TPTrans needs a very long time to converge as the large size pair-wise paths and the sparse scatter algorithm. The leading results demonstrate that our multi-scale formulation for the task of source code summarization is reasonable. To further evaluate the effectiveness of our method, we provide a case study in Appendix D. Moreover, we provide the results on RMSE-Deepcom dataset with error bars to show that the improvement achieved by our method is statistically significant in Table 11 at Appendix B.

**Human evaluation.** As a complement, human evaluation is conducted over the transformer-based methods following the previous works [12, 27, 15]. We invite 5 volunteers with over 5 years of programming experience and ask them to score (0 - 10, 10 is the best) the generated summaries over 100 examples sampled randomly from the testing sets (20 from RMSE-Deepcom and 80 from Fun-com). Here we focus on two key points: **naturalness** (grammaticality and fluency of the generated summary) and **usefulness** (what extent the generated summary is useful to understand the code). We divide the 100 examples into 5 groups (each one has 20 examples) and assign each group into 3

Table 3: Human evaluation results over 100 examples sampled randomly from the testing sets (20 from RMSE-Deepcom and 80 from Funcom).

| Methods | Naturalness | Usefulness |
|---|---|---|
| NeuralCodeSum | 5.12 | 4.98 |
| TPTrans | 5.51 | 5.23 |
| CAST | 5.55 | 5.19 |
| PA-former (ours) | **6.09** | **5.86** |

different volunteers, and the final score of each example is the average of its 3 "review" scores. As shown in Table 3, our PA-former outperforms others in both naturalness and usefulness.

## 4.3 Ablation Study

In this part, we analyze the importance of designs in our proposed PA-former using a series of ablation studies on EMSE-DeepCom dataset. Unless otherwise specified, the training setup is the same as above.

**Multi-scale *v.s.* single-scale.** To demonstrate the effectiveness of multi-granularity formulation, we degenerate the pyramid input into a single large sequence by concatenating the three granularity sequences and directly feed the same sequence into a relative positional encoded Transformer with a copy mechanism. As shown in Tabel 4, more data in the input sequence obtains performance improvement compared to NeuralCodeSum which only takes a sub-token sequence as input. However, processing the same data using our proposed multi-scale way achieves higher performance.

Table 4: Evaluation results of multi-scale *v.s.* single-scale.

| Methods | BLEU | Rouge-L | Meteor | Precision | Recall | F1 |
|---|---|---|---|---|---|---|
| multi-scale | **38.848** | **56.095** | **25.895** | **62.498** | **56.642** | **57.903** |
| single-scale | 37.329 | 55.032 | 25.114 | 61.586 | 55.750 | 56.946 |

**Effectiveness of pyramid attention.** The core components of pyramid attention are the bottom-to-top aggregation and cross-granularity interaction. We show the necessity for both of these two components here. To demonstrate the effectiveness of bottom-to-top aggregation, we evaluate a variant of PA-former whose bottom-to-top pathway is disabled. As shown in Table 5, the model performance drops by about 1.1 BLEU scores without such a component. To demonstrate the effectiveness of cross-granularity interaction, we evaluate a variant of PA-former whose cross-granularity interaction is disabled which directly degenerates into NeuralCodeSum as we only use the fine-grained encoder output for decoding. As shown in Table 5, the model performance drops by about 1.7 BLEU scores without such a component. Such results reveal the importance of cross-granularities interaction. We further disable the granular-aware relative embedding to investigate its importance, and we observe a big drop in performance shown in Table 5, which shows its effectiveness.

Table 5: Evaluation results of pyramid attention ablation.

| Methods | BLEU | Rouge-L | Meteor | Precision | Recall | F1 |
|---|---|---|---|---|---|---|
| PA-former | **38.848** | **56.095** | **25.895** | **62.498** | **56.642** | **57.903** |
| *w/o bottom-to-top* | 37.532 | 55.153 | 25.131 | 61.815 | 55.590 | 56.977 |
| *w/o cross-granularity* | 37.133 | 54.800 | 25.051 | 61.356 | 55.413 | 56.682 |
| *w/o granular-aware rel.* | 36.905 | 54.690 | 24.896 | 61.443 | 55.224 | 56.560 |

**Decoding strategy.** The output of our PA-former encoder is a pyramid representation, thus there are various ways for decoding. Our PA-former directly uses the fine-grained sequence just for simplicity. Here, we study other implementations that utilize all three granularity of representation. The most intuitive method is concating the three granularity representations, and another way is using

them in a serial strategy [18]. As Table 6 shows, using the fine-grained representation is good enough in this task to benefit from the design of information interaction across granularities. The results are a bit counterintuitive, we think a reasonable explanation is that the fine-grained representation already contains enough information benefited from the proposed pyramid attention. When all three granularities are used, there is severe information redundancy between different granularities, which complicates the task.

Table 6: Evaluation results of decoding strategy.

| Decoding strategy | BLEU | Rouge-L | Meteor | Precision | Recall | F1 |
|---|---|---|---|---|---|---|
| fine-grained | **38.848** | **56.095** | **25.895** | **62.498** | **56.642** | **57.903** |
| serial | 38.729 | 55.894 | 25.803 | 62.223 | 56.425 | 57.669 |
| concat | 38.556 | 55.819 | 25.797 | 62.174 | 56.456 | 57.638 |

**Grammar input.** When constructing the pyramid input, we intentionally enforce the bias between different granularities by assigning different types of content. For example, we bias the semi-grained sequence into grammar features by assigning token grammar types to it. To evaluate the rationality of assigning different types of content, we compare the proposed pyramid input without grammar information, which set the initial content of semi-grained and coarse-grained sequences to empty. The results in Table 7 prove the rationality of pyramid input design.

Table 7: Evaluation results of grammar input.

| pyramid input content | BLEU | Rouge-L | Meteor | Precision | Recall | F1 |
|---|---|---|---|---|---|---|
| PA-former | **38.848** | **56.095** | **25.895** | **62.498** | **56.642** | **57.903** |
| *w/o grammar* | 38.394 | 55.744 | 25.706 | 62.198 | 56.458 | 57.637 |

## 5 Conclusion

In this paper, we present a multi-scale formulation for source code summarization. Given a code snippet, we construct a pyramidal input with three granularities. And a novel pyramid attention mechanism is proposed for efficient information interaction across granularities. Without bells and whistles, the proposed method obtains 5% improvement over the strong baseline and achieves new state-of-the-art results. Additional ablation studies show the effectiveness of the proposed modules. Furthermore, our multi-scale formulation for the source code can be easily extended to other code-understanding tasks. In future work, we would try to scale our model up and pre-train it on larger source code corpora to further improve its capability and generalizability on source code comprehension tasks.

## 6 Acknowledgement

This research was supported by NSFC (62076121, 61921006). The authors would like to thank Ying Li, Hao-Yuan He, Ren-Biao Liu, Yu-Xi Sun and Zhi-Yu Shen for helpful discussions. Finally, We are grateful to all reviewers for their insightful comments.

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
