# Appendix

We first provide the implementation details of PA-former and the compared baselines (Appendix A). Then, more supplementary experiments will be listed in Appendix B. Next, we introduce the data process for code (Appendix C). Finally, qualitative examples are provided in Appendix D.

## A  Implementation Details

### A.1  PA-former

**Pyramid attention**   Here we provide the pseudocode of our pyramid attention forward process in a PyTorch-like style in Algorithm 1. Compared with other methods, the algorithm implementation is straightforward and efficient. Please refer to the provided code in `pa_former` for more details.

---
**Algorithm 1** PyTorch-like pseudocode of pyramid attention

---
```
# self_attn: relative position encoded multi-head self-attention module
# ln_*: layer norm layer
# fc_*: linear layer
# fine_seq (BxLfxD): batched fine-grained sub-token sequence
# semi_seq (BxLsxD): batched semi-grained token sequence
# fine_seq (BxLcxD): batched coarse-grained statement sequence
# fine_to_semi (BxLsxLf): mappings between fine_seq and semi_seq
# fine_to_coarse (BxLcxLf): mappings between fine_seq and coarse_seq
# rel_pos_map (BxLxL): relative position map

# normalize mappings
fine_to_semi /= (fine_to_semi.sum(dim=-1, keepdim=True) + 1e-8)
fine_to_coarse /= (fine_to_coarse.sum(dim=-1, keepdim=True) + 1e-8)

# bottom-to-top aggregation process
# here, @: batch matrix multiplication
semi_seq = ln_s(fc_s(fine_to_semi @ fine_seq) + semi_agg)
coarse_seq = ln_c(fc_c(fine_to_coarse @ fine_seq) + coarse_seq)

# cross-granularity interaction
# here, cat: concatenation
x = cat((fine_seq, semi_seq, coarse_seq), dim=1) # BxLxD
x_lens = fine_seq.size(1), semi_seq.size(1), coarse_seq.size(1)
x = self_attn(x, rel_pos_map) # BxLxD
fine_seq, semi_seq, coarse_seq = split(x, x_lens, dim=1)
```
---

### A.2  Baselines

In this section, we provide the implementation details of baselines.

CODE-NN [12] is a basic attention-based seq2seq model which takes a token sequence as input. For the encoder, we use a 2-layer bi-directional LSTM and set the hidden size to 256. For decoder, we use a 2-layer single-directional LSTM and set hidden size to 512. Following [30], the hidden and cell vectors of the bi-directional encoder is concated separately to form 512-D hidden vectors which are used as the initial hidden states for decoder. And the copy mechanism is used over the encoder-decoder framework. This method directly uses the token sequence, which leads to a big risk of out-of-vocabulary and limits the performance of this method.

TreeLSTM [29] is a attention-based tree2seq model which takes a parsed AST as input. For the encoder, we use child-sum tree-based LSTM with the hidden size of 512 and implement it using the excellent Deep Graph Library (DGL) [5]. For the decoder, we use a 2-layer single-directional LSTM and set the hidden size to 512. The hidden and cell vectors of the root node are used as the initial hidden states for decoder. And a copy mechanism is used over the encoder-decoder framework. As

---
[5] https://www.dgl.ai/

the large size of AST and the risk of out-of-vocabulary, such a structure-aware doesn't obtain large performance improvement.

HDeepCom [11] is a hybrid method that encodes the flattened AST sequence and sub-token sequence separately, and combines these two representations when decoding using the attention mechanism. We set the embedding size and hidden size to 512 for a fair comparison.

ASTAttnGRU [13] is also a hybrid method that encodes the AST-based graph and sub-token sequence separately, and combines these two representations when decoding using the attention mechanism. We use the DGL to implement the tree encoder and set the embedding size and hidden size to 512 for a fair comparison.

SiT [35] uses the structure information (abstract syntax edges, control flow edges, and data dependency edges) as attention mask to guide the attention computations. Following the paper, this method only considers the punctuation-removed token sequence. Copy mechanism is applied and the model configuration is consistent with the default Transformer.

GREAT [9] introduces various edge types following [1] and models the pairwise relations using the summed edge type embeddings between two terminal nodes (i.e., token) in AST. Then these relations are used as relative distance bias in the attention computation process as Equation 4. They only consider the token sequence and embed the tokens by averaging embeddings of its sub-token(s). We also add a copy mechanism over the encoder-decoder framework for a fair comparison.

NeuralCodeSum [30] uses the sub-token sequence as input and directly adapts relative position encoded Transformer. And copy mechanism is also introduced in this method.

CAST [27] is a hybrid method that encodes the sub-token sequence using the Transformer encoder and split the AST into statement subtrees for structure representation learning. Then use tree-based RvNN to encode each subtree and then a max-pooling is applied to get the statement-level structure representations. Next, these statement-level representations are combined into a tiny tree according to the AST to capture global structure information. For the decoder, a serial attention mechanism is used to combine the learned sequential and structural features. And copy mechanism is also used in this method.

TPTrans [25] integrates the absolute paths and relative paths between sub-tokens into Transformer by representing the paths as order non-terminal node sequences and encoding them using GRUs. The learned path embeddings are used as a relative relation bias in attention computations as Equation 4. And copy mechanism is also used in this method.

### A.3 Training configuration

We show the details of training configurations in Table 8. The "rnn-24e" is used for all LSTM/GRU-based methods on both middle-size and large-size datasets. The "Transformer-30e" is used for all Transformer-based methods on middle-size dataset. The "Transformer-36e" is used for all Transformer-based methods on large-size dataset. For the items of "lr scheduler", "MS-(12, 19)" means that MultiStep learning rate scheduler which updates the `current_lr` into $0.1\times$ `current_lr` at 12th and 19th epoch.

Table 8: Training configurations.

| tag | #epochs | optimizer | initial lr | lr scheduler | warmup steps | clip grad |
|---|---|---|---|---|---|---|
| rnn-24e | 24 | AdamW | 0.003 | MS-(12, 19) | – | 8.0 |
| Transformer-30e | 30 | AdamW | 0.0002 | MS-(16, 24, 28) | 1500 | 5.0 |
| Transformer-36e | 36 | AdamW | 0.0002 | MS-(16, 24, 28) | 1500 | 5.0 |

## B More Experiment Results

**Improvement for Transformer.** We show the effect of training tricks on improving model performance in Table 9. The results show the optimizer AdamW and warmup strategy help. Such an improvement caused by training tricks also shows the strong ability of the Transformer. And we use these training settings to compare baselines for a fair comparison.

Table 9: Evaluation results of training tricks. "$-$" represents the base model trained using Adam optimizer and MultiStep lr_scheduler which is used in our baseline [30].

| Tricks | BLEU | Rouge-L | Meteor | Precision | Recall | F1 |
|---|---|---|---|---|---|---|
| $-$ | 34.159 | 52.632 | 23.191 | 60.121 | 52.954 | 54.690 |
| + *adamW* | 36.383 | 53.656 | 24.345 | 59.825 | 54.588 | 55.579 |
| + *warmup* | 37.133 | 54.800 | 25.051 | 61.356 | 55.413 | 56.682 |

**Number of encoder layer.** We further perform the ablation study by varying the number of encoder layers and list the results in Table 10. As the results show, a deeper model helps improve performance as more cross-granularity aggregations and interactions bring richer semantic information.

Table 10: Evaluation results of number of encoder layers. #layers represents the number of encoder layers.

| #layers | BLEU | Rouge-L | Meteor | Precision | Recall | F1 |
|---|---|---|---|---|---|---|
| 3 | 37.417 | 54.988 | 25.128 | 61.632 | 55.586 | 56.878 |
| 6 | 38.848 | 56.095 | 25.895 | 62.498 | 56.642 | 57.903 |
| 9 | **39.029** | **56.382** | **26.136** | **62.576** | **57.094** | **58.177** |

**Results with error bar.** We train the models 3 times with different random seeds on RMSE-Deepcom dataset and list the scores with error bars in Table 11. The results suggest that our method achieves significant improvement (e.g. 1.5 on BLEU) over the baselines with low standard deviation ($< 0.1$).

Table 11: Comparisons results with error bars.

| Methods | BLEU | Rouge-L | Meteor | Precision | Recall | F1 |
|---|---|---|---|---|---|---|
| | | | RMSE-Deepcom (middle-size) | | | |
| CODE-NN [12] | $28.414 \pm 0.084$ | $43.463 \pm 0.210$ | $17.869 \pm 0.062$ | $47.404 \pm 0.168$ | $44.922 \pm 0.355$ | $44.715 \pm 0.207$ |
| TreeLSTM [29] | $28.946 \pm 0.157$ | $44.002 \pm 0.117$ | $18.221 \pm 0.086$ | $48.167 \pm 0.318$ | $45.370 \pm 0.232$ | $45.281 \pm 0.130$ |
| HDeepCom [11] | $32.052 \pm 0.180$ | $48.890 \pm 0.197$ | $21.440 \pm 0.124$ | $54.236 \pm 0.052$ | $50.398 \pm 0.262$ | $50.618 \pm 0.193$ |
| ASTAttnGRU [13] | $32.865 \pm 0.159$ | $49.624 \pm 0.119$ | $22.095 \pm 0.107$ | $54.585 \pm 0.232$ | $51.452 \pm 0.340$ | $51.331 \pm 0.118$ |
| SiT [35] | $35.638 \pm 0.048$ | $53.666 \pm 0.082$ | $24.134 \pm 0.065$ | $60.566 \pm 0.202$ | $54.269 \pm 0.073$ | $55.637 \pm 0.100$ |
| GREAT [9] | $36.388 \pm 0.086$ | $53.665 \pm 0.062$ | $24.193 \pm 0.060$ | $60.176 \pm 0.193$ | $54.266 \pm 0.092$ | $55.545 \pm 0.115$ |
| NeuralCodeSum [30] | $37.044 \pm 0.098$ | $54.653 \pm 0.179$ | $24.918 \pm 0.126$ | $61.193 \pm 0.140$ | $55.301 \pm 0.181$ | $56.541 \pm 0.154$ |
| CAST [27] | $37.089 \pm 0.106$ | $54.794 \pm 0.075$ | $24.990 \pm 0.079$ | $61.452 \pm 0.151$ | $55.346 \pm 0.023$ | $56.668 \pm 0.079$ |
| TPTrans [25] | $37.225 \pm 0.023$ | $54.954 \pm 0.043$ | $24.993 \pm 0.029$ | $61.739 \pm 0.283$ | $55.433 \pm 0.079$ | $56.828 \pm 0.056$ |
| PA-former (ours) | $\mathbf{38.777} \pm 0.093$ | $\mathbf{56.071} \pm 0.069$ | $\mathbf{25.901} \pm 0.016$ | $\mathbf{62.489} \pm 0.168$ | $\mathbf{56.650} \pm 0.031$ | $\mathbf{57.908} \pm 0.077$ |

## C  Data preprocessing

For the summaries of code which are natural language sentences, we preprocess them following the natural language process (NLP) pipeline. In this section, we mainly focus on the code preprocessing pipelines which are based on the parsed AST. Given a code snippet, we use Tree-sitter to get the parsed AST, an example is illustrated in Figure 5.

### C.1  Pyramid input

We show the details of our proposed pyramid input construction with the example in Figure 5.

The fine-grained sequence $\mathcal{T}$ is easy to handle. We first extract the ordered terminal nodes from AST, we get [public, int, parseStringAsInt, (, String, s, ), ...] for the given example. And then we split each element in the token sequence into sub-tokens and we get $\mathcal{T} =$ [public, int, parse, String, As, Int, (, String, ...]. Here, we ignore the grammatical information and just treat elements of $\mathcal{T}$ as natural words, so we name it as $\mathcal{T}$ext sequence.

We use the semi-grained sequence $\mathcal{G}$ to represent the $\mathcal{G}$rammatical details of the code, which is a complement to $\mathcal{T}$. Each element in $\mathcal{G}$ corresponds to an element in the token sequence. For

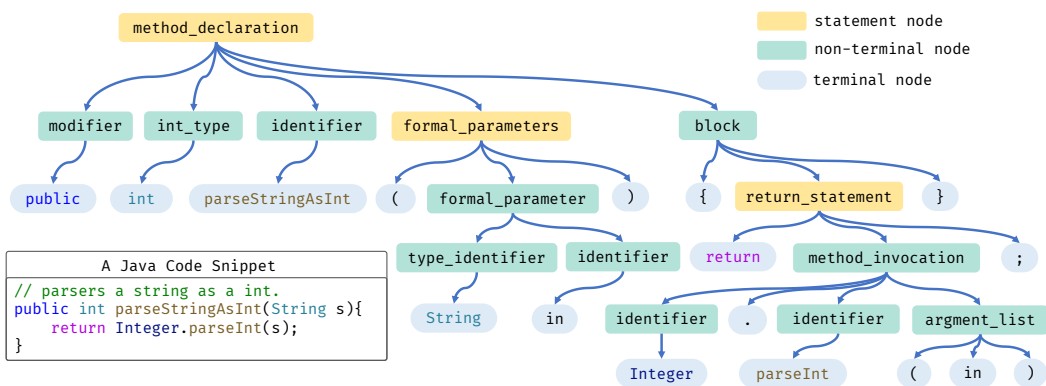

Figure 5: Parsed abstract syntax tree (AST) example using Tree-sitter.

each token, its parent node in AST represents its grammatical property, e.g., `parseStringAsInt` is an `identifier` and `public` is a `modifier`. However, we notice that all variable names and method names are marked as `identifier`. To further differentiate them, we instead use their grandparents' node in ASTs to mark the grammatical types, e.g. `parseStringAsInt` is marked as `method_declaration` and `parseInt` is marked as `method_invocation`. Finally, we get $\mathcal{G}$ = [modifiers, int_type, method_declaration, (, type_identifier, ...]. For the mapping relations $\mathcal{M}_{\mathcal{G}\rightarrow\mathcal{T}} \in \mathbb{R}^{|\mathcal{G}|\times|\overline{\mathcal{T}}|}$, we set $\mathcal{M}_{\mathcal{G}\rightarrow\mathcal{T}}(i,j)$ to 1 if $\mathcal{T}[j]$ is split from the token corresponding to $\mathcal{G}[i]$. In this example, we set $\mathcal{M}_{\mathcal{G}\rightarrow\mathcal{T}}(3,3), \mathcal{M}_{\mathcal{G}\rightarrow\mathcal{T}}(3,4), \mathcal{M}_{\mathcal{G}\rightarrow\mathcal{T}}(3,5)$ and $\mathcal{M}_{\mathcal{G}\rightarrow\mathcal{T}}(3,6)$ to 1, as [parse, String, As, Int] is split from `parseStringAsInt`.

The $\mathcal{L}$ogical nature of the code is mostly in statement-level coarse-grained sequence $\mathcal{L}$. As the type of statement is limited, we first set the pre-defined statement set $\mathcal{S}$ shown in Table 12 for java dataset. In addition to the normal java statements, we also treat `method_declaration` and `formal_parameters` as statements. We treat the node in AST as a statement node, if its tag is in $\mathcal{S}$. Thus we get $\mathcal{L}$ = [method_declaration, formal_parameters, return_statement] (yellow nodes in Figure 5). For the mapping relations $\mathcal{M}_{\mathcal{L}\rightarrow\mathcal{G}} \in \mathbb{R}^{|\mathcal{L}|\times|\mathcal{G}|}$, we set $\mathcal{M}_{\mathcal{L}\rightarrow\mathcal{G}}(i,j)$ to 1 if $\mathcal{L}[i]$ is the nearest ancester of $\mathcal{G}[j]$ in the scope of $\mathcal{L}$. In this example, we set $\mathcal{M}_{\mathcal{L}\rightarrow\mathcal{G}}(2,4), \mathcal{M}_{\mathcal{L}\rightarrow\mathcal{G}}(2,5), \mathcal{M}_{\mathcal{L}\rightarrow\mathcal{G}}(2,6)$ and $\mathcal{M}_{\mathcal{L}\rightarrow\mathcal{G}}(2,7)$ to 1, as `formal_parameters` is the nearest ancester of ( , String, in and ).

Table 12: The pre-defined statement set.

| statement type | tag of statement node |
| --- | --- |
| expression statement | `expression_statement, explicit_constructor_invocation, local_variable_declaration` |
| exit statement | `return_statement, yield_statement, throw_statement` |
| conditional statement | `if_statement, switch_expression` |
| loop statement | `for_statement, enhanced_for_statement, while_statement do_statement, continue_statement, break_statement` |
| exception statement | `try_statement, catch_clause, finally_clause` |
| others | `method_declaration, formal_parameters` |

## C.2 Data preprocessing for baselines

Since some compared baselines don't provide available preprocessing code, we have to implement the preprocessing pipelines according to their papers.

For GREAT [9], we add the additional edges to AST according to construct the code graph representation according to [1]. The edges include `NextToken, LastRead, LastWrite, ComputeFrom`

LexicalUse and LeafCFG. The edge LeafCFG represents the edge of control flow graph (CFG) (see details in [9]) and please refer to [1] for details about other edges.

For SiT [35], we add Flow edge between each in-statement token pair, add adopt LexicalUse, LastRead and LastWrite in GREAT into the Data dependency edge.

## D   Qualitative Examples

In this section, we provide several samples of code summarization with different models. We can see that PA-former generates the most precise comment in most cases.

```java
public boolean isOnPieChart(Point screenPoint){
    double sqValue = (MathUtil.pow(mCenterX - screenPoint.getX(), 2)\
        + MathUtil.pow(mCenterY - screenPoint.getY(), 2));
    double radiusSquared = mPieChartRadius * mPieChartRadius;
    boolean isOnPieChart = sqValue <= radiusSquared;
    return isOnPieChart;
}
```
Reference: checks if point falls within piechart.
PA-former (ours): checks if a point falls within a pie chart.
TPTrans: determines if a point is on a pie chart.
CAST: determines if a point is on the pie chart.
NeuralCodeSum: returns true if the specified point is on the pie chart.
GREAT: checks if point is within pie pie pie pie pie pie pie area.
SiT: checks if a point is on the chart.
ASTAttnGRU: checks if at the current position.

```java
public XmlReporter(OutputStream out, String title){
    this.out = out;
    this.title = title;
}
```
Reference: create xmlreporter which writes to the given stream.
PA-former (ours): create an xml reporter which writes to the given stream.
TPTrans: creates a new xml reporter.
CAST: create an xml reporter to write to the given stream.
NeuralCodeSum: create an instance of this class.
GREAT: create an xml stream.
SiT: create a new reporter.
ASTAttnGRU: create an object decoding the values from the passed stream.

```java
public final boolean isModified(boolean isEnable) {
    if(isStopped()) {
        return flase;
    }
    DependencyContainer dependencies = _dependencies;
    if (dependencies == null) {
        return false;
    }
    if (isEnable) {
        boolean isModified = dependencies.isModified();
        return isModified;
    } else {
        boolean isModified = isModified(getParent());
        return isModified;
    }
}
```
Reference: returns true if any of the classes have been modified.
PA-former (ours): returns true if any of the dependencies have been modified.
TPTrans: returns true if the underlying dependencies have changed.
CAST: returns true if the configuration of this object has changed.
NeuralCodeSum: returns true if the user has changed the dependency container.
GREAT: returns true if the entry has been modified.
SiT: returns true if the underlying dependencies have changed.
ASTAttnGRU: returns whether or not this property has been modified or not.

```java
public int consumeInteger()
    throws ParsingException , IllegalStateException {
    if (completed) {
        throwNoMoreContent();
    }
    String value = currentToken().value();
    try {
        int result = Integer.parseInt(value);
        moveToNextToken();
        return result;
    } catch ( NumberFormatException e ) {
        Position position = currentToken().position();
        throw new ParsingException(position, value);
    }
}
```

Reference: convert the value of this token to an integer return it and move to the next token.
PA-former (ours): convert the value of this token to an integer and move it to the next token.
TPTrans: consume an integer value from the input stream.
CAST: consume an integer value off the current token throwing an exception if it is not an integer.
NeuralCodeSum: gets the next token and remove it from the list.
GREAT: convert this token to an integer and move to an integer.
SiT: consume an integer value from the request.
ASTAttnGRU: gets the value of the int property.

```java
public void delete(String path, ParameterMap params,
    AsyncCallback callback) {
        HttpDelete req = new HttpDelete(path, params);
        executeAsync(req, callback);
}
```

Reference: execute a delete request and invoke the callback on completion.
PA-former (ours): execute a delete request and invoke the callback on completion.
TPTrans: execute a delete request.
CAST: execute a delete request.
NeuralCodeSum: execute a delete delete request.
GREAT: perform a delete on the specified path.
SiT: execute a delete request.
ASTAttnGRU: delete a parameter from a map.

```java
public FXDialog(final Stage parent , final Parent content, final
    Modality modality, final double width , final double height) {
    init(parent, modality, content);
    dialog.setWidth(width);
    dialog.setHeight(height);
    final Scene scene = new Scene(content, width, height);
    setScene (scene);
}
```

Reference: create a new dialog with a specified width and height.
PA-former (ours): create new dialog with the specified width and height.
TPTrans: creates a new dialog.
CAST: create dialog with parent content.
NeuralCodeSum: initializes the dialog with the given parent modality and content.
GREAT: creates new form <unk>.
SiT: creates a new dialog box.
ASTAttnGRU: create a new dialog box.