# OpenReview forum: "Pyramid Attention For Source Code Summarization"
_NeurIPS.cc/2022/Conference — NeurIPS 2022 Accept_

### Official Review · Reviewer_Xtyo · 2022-07-08

**Rating:** 7
**Confidence:** 4
**Soundness:** 3 good
**Presentation:** 3 good
**Contribution:** 3 good

**Summary:**

The paper presents the method to tackle the source code summarization task by leveraging multi-scale features that appear in it. Such a human-like multi-scale comprehension is modeled relatively easily using the Transformer architecture enhanced with the information at each granularity level and cross-granularity interactions.

**Questions:**

1. How much longer are inputs being passed to the multi-head cross-attention?
1a. How much did the computational cost increase due to this?
2. I wish the paper provided the results with error bars. Is it possible to add this?
3. How was the segmentation on each level performed? Were there any additional tools used for such a segmentation? Is the description of "Data preprocessing" section enough?

**Limitations:**

Authors do not provide limitation section nor analyze the results qualitatively. I believe one important factor that would be fair to disclose is the computational expense of the model, relative to the Vanilla Transformer baseline.

**Strengths And Weaknesses:**

*Pros*:
1. I like how the paper is structured and how illustrations benefit the narrative. A reader gets a nice overview before digging into details (Section 3.1 provides details on something that was previously elegantly explained on a high level, making it easy to process)
2. Design choices are motivated well (e.g., use of the vocabulary, graining of the sequence, cross-granular interactions). Important ablations are performed; in general, paper naturally answers most questions that a reader wants to ask.
3. The proposed method is elegant, does not overcomplicate the process, and seems easily reproducible. Moreover, the results suggest it provides stable gains on both used datasets.

*Cons*:
1. Since page 6, the quality of grammar and writing radically decreases, suggesting it needs proofreading, as, e.g., "All model is trained using..." is one among other trivial mistakes.
2. A bit of obscured equations (4)-(9) suddenly makes processing this part much harder than the rest of the paper. Interestingly, this gracious note made me understand the steps easily: "we just need concat the three sequences and the relative position maps, then directly adopt a relative position encoded multi-head self-attention layer to implement the cross-granularity interaction module." I recommend reworking the "Cross-granularity interaction" section a bit and, again, allowing someone to proofread that.

---

> ### Author Response · Authors · 2022-08-02
> **Input and computational cost; Error bars; Multi-scale segmentation method**
>
> Thanks a lot for your insightful comments and valuable suggestions for improvements on our paper! We are glad that you like the proposed simple yet efficient approach and our paper construction and illustrations. Below, we address your questions:
>
> **Input and computational cost**: In fact, we provided the input size in tabel 1. We re-list the input length here:
>
> | Dataset | max fine-seq length|max semi-seq length|max coarse-seq length|
> |:-------------:|:-----------:|:----------:|:------:|
> |EMSE-Deepcom |196|160|16|
> |FunCom|256|196|32|
>
> Compared with the methods that only use sub-token sequences (like NeuralCodeSum), our length is 1.9 time longer, which makes the attention computation increase about 3.56 times. We think how to reduce the computation cost may be a very potential research direction.
>
> **Error bars**: Yes! We actually run serval times with different random seeds and list the median scores in Table 2. Here we provide part of the evaluation results with error bars over strong methods like CAST and TPTrans on RMSE-Deepcom dataset (full version will be added to Appendix):
>
> | Methods              |  Bleu  | Rouge-L | Meteor | Precision | recall  |   f1    |
> |:---------------------|:------:|:-------:|:------:|:---------:|:-------:|:-------:|
> | NeuralCodeSum        | 37.044 $\pm$ 0.098 | 54.6528 $\pm$ 0.179 |	24.918 $\pm$ 0.126 | 61.193 $\pm$ 0.140 | 55.301 $\pm$ 0.181 |	56.541 $\pm$ 0.154 |
> | CAST                 | 37.089	$\pm$ 0.106 | 54.7935 $\pm$	0.075 | 24.990 $\pm$ 0.079 |61.4505 $\pm$ 0.151 | 55.346 $\pm$ 0.023 | 56.668 $\pm$ 0.079 |
> | TPTrans              | 37.225 $\pm$ 0.023 | 54.9535 $\pm$ 0.043 | 24.993 $\pm$ 0.029 |61.7395 $\pm$ 0.283 | 55.433 $\pm$ 0.079 | 56.828 $\pm$ 0.056 |
> | PA-former (ours)     | 38.777 $\pm$ 0.093 | 56.071 $\pm$ 0.069 | 25.901 $\pm$ 0.0164 |62.489 $\pm$ 0.168 | 56.650 $\pm$ 0.031 | 57.908 $\pm$ 0.077 |
>
> **Multi-scale segmentation method**: As the "Data Preprocessing" section says, we use the excellent tree-sitter as the AST parser, and do the segmentation following the descriptions in section "Pyramid Input Constructor". What's more, a detailed data-processing example is provided in Appendix C. And we will open-source our code which includes the data processing scripts if the paper is accepted.
>
> What's more, we apologize for the typos and errors in our writing as the time constraints. And thank a lot for your kind corrections and suggestions! We will fix them and improve the writing in the final revision.

---

> > ### Comment · Reviewer_Xtyo · 2022-08-08
> > **Thank you for the answer**
> >
> > I want to thank the authors for their answers. I will keep my decision to recommend the paper for acceptance firmly.
> >
> > When considering appropriate error bars, the results suggest the method achieves quite a high improvement over the baselines. Specifically, in BLEU(+1.5), Rouge-L(+1.0), and F1 metrics(+1.0), Meteor(+0.9) there is a notable improvement, while the standard deviation is lower than 0.1. This indicates the proposed approach outperformed previous strong methods by at least ten standard deviations! It is an substantial improvement, and from the statistical point of view, a significant one.
> >
> > Additionally, the authors answered my concerns regarding the quality of writing in the latter part of the paper. I believe that the authors will deliver such improvement quickly, given that, e.g., the introduction, method description, and structure are well planned and well written, the narrative is easy to follow, and the pictures are prepared with great care.
> >
> > I also read other reviewers' perspectives on the paper and do not find the mentioned weaknesses to be of such great importance. Alternatively, in some cases, I find them to be answered comprehensively by authors, e.g., when the reviewer *26RW* touched on the implications of automatic evaluation and insignificance of the results, the authors provided error bars to prove they outperformed previous SOTA by a giant margin and, additionally, presented human evaluation to confirm the results of automatic evaluation.

---

> > > ### Author Response · Authors · 2022-08-09
> > > **Thanks**
> > >
> > > Thank you for the appreciation of our work, it gives us a lot of confidence! And we will improve the quality of writing as soon as possible.

---

### Official Review · Reviewer_26RW · 2022-07-09

**Rating:** 4
**Confidence:** 4
**Soundness:** 3 good
**Presentation:** 3 good
**Contribution:** 3 good

**Summary:**

This paper proposes a multi-granularity method for the code summarization task. Specially, one code snippet is first hierarchically broken into three granularities: statements, tokens, and sub-tokens levels. Then a new mechanism named pyramid attention mechanism is proposed for the aggregation of multi-granularity features. The authors also conducted extensive experiments to demonstrate the performance of their model and the effectiveness of the model components.


**Questions:**

1. Are the improvements achieved by the proposed method over existing work significant?

2. How did you calculate the BLEU scores of your approach and the baselines in Tables 2-6?


**Limitations:**

No, the authors did not adequately addressed the limitations and potential negative societal impact of their work.

**Strengths And Weaknesses:**

This paper targets automatic generation of natural language descriptions for source code, which could be useful in a variety of source code related tasks.  The paper illustrates and explains the idea well. The paper is generally well organized and clearly describes the architecture of the model. The proposed multi-granularity method for code representation is novel to me.

The improvement of the proposed method over related work is rather small. Although the paper says that “the proposed methods obtain 5% improvement over the strong NeuralCodeSum on a middle-size dataset and achieve new state-of-the-art results”, actually the results achieved by the proposed method are very similar to the results of CAST and TPTrans, as shown in Table 2. The differences are around or below 1.0 in almost all metrics. It is not clear if those improvements are indeed significant.

Also, there is lack of human evaluation to show the effectiveness of the proposed method in practice. Automatic evaluation (using BLEU metric, etc) may not be sufficient. Some papers (e.g. [a] and [b] below) on code summarization show that automatic metrics mainly calculate the textual similarity between the reference and the generated summaries, rather than the semantic similarity. Thus, human evaluation is needed to confirm the effectiveness of a code summarization method in practice.

[a] Li, J., Li, Y., Li, G., Hu, X., Xia, X., & Jin, Z., EDITSUM : A Retrieve-and-Edit Framework for Source Code Summarization.  Proc. ASE2021.

[b]J. Zhang, X. Wang, H. Zhang, H. Sun, and X. Liu, “Retrieval-based Neural Source Code Summarization.” Proc. ICSE2020.

Furthermore, the below reference [c] shows that the BLEU metric, which is widely used in existing code summarization work, actually has many variants. Ignoring the differences among these variants could greatly affect the validity of the claimed results. Please check the BLEU scores in Tables 2-6 and make sure that they are calculated in the same way.

[c] Shi, E., et al., On the Evaluation of Neural Code Summarization, in Proc.  ICSE2022.

Minor:

. The reference [26] is wrong. CAST is published at EMNLP2021 instead of ICPC2021.

. Line 259. Bleu -> BLEU. Please keep the term consistent in the paper.

---

> ### Author Response · Authors · 2022-08-02
> **Significant improvements; BLEU score calculation; Human evaluation;**
>
> Thanks a lot for your review! We answer your questions below:
>
> **Significant improvements**: We think our method does have significant improvements which should be measured in a relative viewpoint. Recent SOTA methods like TPTrans and CAST both based on NeuralCodeSum (the baseline) which directly uses transformer for code summarization task. As the Table 2 shows, our method gains more improvement when evaluated using the same training settings (BLEU imporvement to NeuralCodeSum: 1.7 v.s. 0.7). Note that for the sake of fairness we uniformly use a stronger training strategy for all methods, see Appendix A.3 for details.
>
> Furthermore, we provide part of the evaluation results with error bars over strong methods like CAST and TPTrans on RMSE-Deepcom dataset below. The results suggest the our method achieves significant improvement (e.g. 1.5 on BLEU) over the baselines with low standard deviation (<0.1).
>
> | Methods              |  Bleu  | Rouge-L | Meteor | Precision | recall  |   f1    |
> |:---------------------|:------:|:-------:|:------:|:---------:|:-------:|:-------:|
> | NeuralCodeSum        | 37.044 $\pm$ 0.098 | 54.6528 $\pm$ 0.179 |	24.918 $\pm$ 0.126 | 61.193 $\pm$ 0.140 | 55.301 $\pm$ 0.181 |	56.541 $\pm$ 0.154 |
> | CAST                 | 37.089	$\pm$ 0.106 | 54.7935 $\pm$	0.075 | 24.990 $\pm$ 0.079 |61.4505 $\pm$ 0.151 | 55.346 $\pm$ 0.023 | 56.668 $\pm$ 0.079 |
> | TPTrans              | 37.225 $\pm$ 0.023 | 54.9535 $\pm$ 0.043 | 24.993 $\pm$ 0.029 |61.7395 $\pm$ 0.283 | 55.433 $\pm$ 0.079 | 56.828 $\pm$ 0.056 |
> | PA-former (ours)     | 38.777 $\pm$ 0.093 | 56.071 $\pm$ 0.069 | 25.901 $\pm$ 0.0164 |62.489 $\pm$ 0.168 | 56.650 $\pm$ 0.031 | 57.908 $\pm$ 0.077 |
>
> **BLEU score calculation**: We already provided our source code in *supplementary material*, your can find the BLEU calculation in "supp_root/pa_former/evaluation/bleu". Actually, we adopted the evaluation code from NeuralCodeSum([https://github.com/wasiahmad/NeuralCodeSum](https://github.com/wasiahmad/NeuralCodeSum)). All the evaluation scores are calculated in the same way.
>
> **Human evaluation**: We agree that the automatic evaluation (like BLEU) may lead to biased results. But we have to admit that automatic evaluation is still the most objective way of evaluating model performance. The reason why we did not perform human evaluation is mainly that we lack a standard process and objective evaluation criteria. Instead, we provide several qualitative examples to demonstrate that our PA-former generates the most precise comment in most cases in Appendix D.
>
> As a complement, we do a preliminary human evaluation over the transformer-based methods with 5 volunteers with over 5 years of programming experience in the past few days. We asked them to score (0 - 10, 10 is the best) the generated summaries of the compared methods over 100 examples sampled randomly from the Java dataset. We list the results below. The human evaluation results and more details will be added to Appendix in our final revision.
>
> | Methods       | Naturalness | Usefulness |
> |:-------------:|:-----------:|:----------:|
> | NeuralCodeSum | 5.12        | 4.98       |
> | TPTrans       | 5.51        | 5.23       |
> | CAST          | 5.55        | 5.19       |
> | PA-former (ours)    | 6.09        | 5.86       |

---

### Official Review · Reviewer_Hb7s · 2022-07-09

**Rating:** 7
**Confidence:** 4
**Soundness:** 3 good
**Presentation:** 2 fair
**Contribution:** 3 good

**Summary:**

The paper introduces pyramid attention for source code summarization and
integrates it into a transformer architecture (they call the resulting
transformer PA-former). The pyramid attention combines three levels of looking
at the source code (statement, token, and subtoken level). The levels are
combined bottom up from subtoken to statement level, and also with a
cross-granularity interaction across levels where each level can query every
other level (including its own).


**Questions:**

Table 5 shows that using just the fine-grained representation for decoding
slightly outperforms using all three representations. I find this
counterintuitive. Could you explain this finding?

In Table 6, the difference between PA-former and w/o grammar is very low. Could
it mean that the input design (e.g., the token grammar types) is less important,
and the good results are obtained because of the pyramid attention and the
structure the input provides for it ($M_{G \rightarrow T}$ and $M_{L \rightarrow
G}$?


**Limitations:**

I cannot think of any limitations which are not addressed.


**Strengths And Weaknesses:**

## Strengths

I think that the presented method is elegant, logical, and not overcomplicated.
It also improves state-of-the-art results.

## Weaknesses

I think the writing could be improved. Overall the paper is written well and is
easy to follow, but there are numerous typos and errors. I found Section 3.1 a
little bit hard to understand at first reading, especially the "Semi-grained
sequence" part, the writing could be much improved there. Correcting the typos
and the plurals would help, and maybe a figure about the AST like Figure 5 in
the Appendix and concrete examples if there is space for that.

There is also a typo in Eq. (2), the disjunction should go over k instead of j.

Some examples of typos and errors:
- line 19: as => because of
- line 124 and 125:  order=> ordered, sequenc => sequence, denotes => denote
- lines 131 and 132: corresponds => corresponding, reveals logic side of program => reveals the logic side of the program.A
- line 172: lenght => length
- line 290: We => we
- line 304: The => the

---

> ### Author Response · Authors · 2022-08-02
> **Decoding using fine-grained representation; The grammar input design**
>
> First of all, we are extremely grateful that you invested so much of your valuable time in reading and providing feedback for our paper. We respond to the concerns and questions raised by you below:
>
> **Decoding using fine-grained representation**: Benefited from the efficient information interaction  of the proposed Pyramid Attention mechanism, the final outputs of the encoder are code representations with three different granularities. We think the fine-grained representation already contains enough information for the summary generation. When using all three granularities, there is serious information redundancy between different granularities. This may make the cross-attention mechanism confused to pay correct attention to each element. As the hierarchy in the multi-granularity representations, may a hierarchy-aware decoder is needed, and this is a good extension for our PA-former.
>
> **The grammar input design**: The usage of grammar is just a small design choice and the ablation study shows that this design is effective with a 0.5 BLEU score gain. And we believe that the importance of the grammar input depends on the task type. The grammar input design can bring more performance improvement for the grammar-focused task like code re-pair, while is less important for this semantic-focused task like code summarization.
>
> And we apologize for the typos and errors in our writing as the time constraints, and thanks a lot for pointing them out. We will fix them and improve the writing in the final revision.

---

> > ### Comment · Reviewer_Hb7s · 2022-08-07
> > **Thank you for your thoughtful answers**
> >
> > Thank you, I increased my score to Accept.

---

### Official Review · Reviewer_Q2Zz · 2022-07-21

**Rating:** 4
**Confidence:** 3
**Soundness:** 2 fair
**Presentation:** 2 fair
**Contribution:** 2 fair

**Summary:**

The proposed idea presented in this work is based on the observation that skilled programmers write and read source codes in a hierarchical way and pay close attention to the conceptual entities like statements, tokens, sub-tokens, and the mapping relations between them. Driven by this observation, the authors argue that a multi-granularities formulation incorporating the entities in different granularities may benefit the code summarization task. To this end, the paper proposes constructing a pyramid-shaped input representation based on a pyramid attention mechanism for efficient feature aggregation and distribution across different hierarchies. The proposed framework is evaluated on two source code summarization benchmarks where it surpasses the prior works and reaches new state-of-the-art results.

**Questions:**

- Are the experiments done with multiple seeds? Are the improvements achieved by PA-former is significant? I just want to make sure that the improvements are not due to some randomness in the experiments.

- Can we apply the proposed framework for other languages, e.g., python, go, PHP? Performing experiments on the CodeXGlue summarization benchmark may add value to the work.

- Is the proposed idea scalable? Can we apply the idea to fine-tune a large language model?

- Figure 1 shows that PA-former generates a better summary by producing additional useful information. But a comparison with ground truth, especially based on BLEU score-like metric, should not be able to demonstrate the efficacy of the proposed approach. What do the authors think about it?

- The paper shows improvements on the target task. But does the paper show whether the proposed model truly learns a pyramid-shaped input representation? Is there any way the authors can validate the claim?



**Limitations:**

My concern is around the authors' observation that skilled programmers write and read source codes in a hierarchical way and pay close attention to the conceptual entities like statements, tokens, sub-tokens, and the mapping relations between them. I am skeptical about accepting this argument. As Figure 1 shows, the PA-former generates a summary with more information, but the reference (human-written summary) does not have the extra information "if the string can be parsed as an integer". Therefore, I felt the observation of the authors is not well supported. Performing a human study to demonstrate that the proposed model-generated summaries are of higher quality would be a plus for this work.

**Strengths And Weaknesses:**

**Strengths**
Generally, the paper did a good job of presenting the motivation, describing the proposed method, and demonstrating the effectiveness of the proposed approach based on experiment and evaluation.

**Weaknesses**

- The paper argues the existing structure-aware code summarization methods are inconsistent with the multi-grained nature of source code. Then the paper linked human behavior to the multi-granularity formulation of source code which I am skeptical to accept. Consequently, it is not clear why the multi-granularity nature of source code would encourage programmers to write summaries accordingly. The paper should verify the main claim of the work.
- Evaluation is only in one language.

---

> ### Author Response · Authors · 2022-08-02
> **Randomness in the experiments; Applied to other languages; Scalability of the idea; BLEU score-like metric; Learning the pyramid-shaped input representation**
>
> Thank you for providing many helpful remarks! Below we answer your questions:
>
> **Randomness in the experiments**: Actually, we did experiments with multiple seeds and provide the evaluation scores with error bars, here we provide part of the evaluation results with error bars over strong methods like CAST and TPTrans on RMSE-Deepcom dataset (full version will be added to Appendix):
>
> | Methods| Bleu|Rouge-L|Meteor|Precision|recall|f1|
> |:---------------------|:------:|:-------:|:------:|:---------:|:-------:|:-------:|
> | NeuralCodeSum| 37.044 $\pm$ 0.098 | 54.6528 $\pm$ 0.179|24.918 $\pm$ 0.126 | 61.193 $\pm$ 0.140 | 55.301 $\pm$ 0.181 |	56.541 $\pm$ 0.154 |
> | CAST  | 37.089	$\pm$ 0.106 | 54.7935 $\pm$	0.075 |24.990 $\pm$ 0.079 |61.4505 $\pm$ 0.151 | 55.346 $\pm$ 0.023 | 56.668 $\pm$ 0.079 |
> | TPTrans | 37.225 $\pm$ 0.023 | 54.9535 $\pm$ 0.043 | 24.993 $\pm$ 0.029 |61.7395 $\pm$ 0.283 | 55.433 $\pm$ 0.079 | 56.828 $\pm$ 0.056 |
> | PA-former (ours) | 38.777 $\pm$ 0.093 | 56.071 $\pm$ 0.069 | 25.901 $\pm$ 0.0164 |62.489 $\pm$ 0.168 | 56.650 $\pm$ 0.031 | 57.908 $\pm$ 0.077 |
>
> For the sake of "significant", we think our improvement is significant enough. Compared with the recent works (TPTrans and CAST) based on NCS, our method achieve the highest improvement. Taking the BLEU score as an example, our method achieves about 1.7 absolute improvement comparing to the 0.7 improvement by TPTrans and CAST.
>
> **Applied to other languages**: Yes! All of the programming language share the hierarchy natures and our method is language-agnostic. The only difficulty is that you need to build the multi-granularity representation yourself following section "Pyramid Input Constructor", because the details of the syntax vary from language to language.
>
> And we think evaluating one programming language is enough. We just claim the multi-granularity formulation is helpful for code modeling and the evaluation of two datasets shows its effectiveness. Actually, we also evaluated our baselines in the python dataset of CodeXGlue benchmark, but we found the dataset is very dirty, and the evaluation scores of both methods are very low (table below). So we think that this dataset doesn't suit code summarization evaluation.  And we argue that there still needs great effort for high-quality datasets in the code comprehension domain, but this is another topic beyond the scope of this article.
> | Methods              |  Bleu  | Rouge-L | Meteor | Precision | recall  |   f1    |
> |:---------------------|:------:|:-------:|:------:|:---------:|:-------:|:-------:|
> | Code-NN | 14.457 | 27.634  | 8.339  | 36.978    | 27.456  | 29.473  |
> | NeuralCodeSum| 17.989 | 37.432  | 13.765 | 49.176    | 37.848  | 40.263  |
>
> **Scalability of the idea**: Yes! the core ideas behind our method are the multi-granularity feature aggregation and information interaction which can be applied to various domains. And our implementation is quite simple based on transformer, we just need to add a feature aggregation module (which can be implemented as a matrix multiplication) and modify the position embedding. Thus, the idea can be applied to big transformer-based language models easily. What's more, we show that deeper model helps improve the performance in Table 9 in Appendix B.
>
> **BLEU score-like metric**: We just use the case in Figure 1 as an example (because its simplicity) to demonstrate that our model can tell richer information for the given code thanks to the multi-granularity formulation. And most of the references in the dataset are informative enough. See the "Qualitative Examples" section in Appendix D for the reference quality and model prediction comparisons.
>
> We agree that the automatic evaluation (like BLEU) may lead to biased results. As a complement, we do a preliminary human evaluation over the transformer-based methods with 5 volunteers with over 5 years of programming experience in the past few days. We asked them to score (0 - 10, 10 is the best) the generated summaries of the compared methods over 100 examples sampled randomly from the Java dataset. We list the results below. The human evaluation results and more details will be added to Appendix in our final revision.
> |Methods|Naturalness|Usefulness|
> |:--:|:--:|:---:|
> |NeuralCodeSum|5.12|4.98|
> |TPTrans|5.51|5.23|
> |CAST| 5.55 | 5.19 |
> |PA-former (ours)| 6.09| 5.86 |
>
> **Learning the pyramid-shaped input representation**: we have done a lot of ablation studies in the section "Ablation Study", the results in table 3 and table 4  demonstrate the effectiveness of multi-granularity formulation and pyramid attention. And we provide "Qualitative Examples" section in Appendix D to show that our PA-former does generate more precise comment in most cases.

---

> > ### Author Response · Authors · 2022-08-09
> > **Concern about the observation of hierarchy in program**
> >
> > Here we continue to answer your questions as the character limitation:
> >
> > **Learning the pyramid-shaped input representation**: Here is an example to show our good summary generation capability of the proposed PA-former:
> > ```
> > public void delete(String path, ParameterMap params, AsyncCallback callback) {
> >     HttpDelete req = new HttpDelete(path, params);
> >     executeAsync(req, callback);
> > }
> > Reference: execute a delete request and invoke the callback on completion.
> > PA-former (ours): execute a delete request and invoke the callback on completion.
> > TPTrans: execute a delete request.
> > CAST: execute a delete request.
> > NeuralCodeSum: execute a delete delete request.
> > ```
> >
> > **Concern about the observation of hierarchy in program**: We explain our motivation for the hierarchical and multi-granularity formulation of source code modeling using a very detailed example in section "Introduction" and Figure 1. And the remaining three reviewers all agree that our paper **illustrates and explains the idea well**. As for the example, we just want to show that our model can tell richer information for the given code, and the generated _"if the string can be parsed as an integer"_ reflects that our model has indeed learned structural information from data thanks to the multi-granularity formulation.  While the relative "bad" _reference (human-written summary)_, it doesn't matter here.

---

> > ### Comment · Reviewer_Q2Zz · 2022-08-09
> > **Author response is not helpful**
> >
> > Thank you for answering my questions/concerns. However, I disagree with a few comments from the authors.
> >
> > - "we think evaluating one programming language is enough."
> > - "we found the dataset (CodeXGlue) is very dirty" - only because the code summarization performance is low does not mean the dataset is dirty. We can argue about the quality of the DeepCom and FunCom datasets too. For example, information leakage across the splits could make models perform better in these datasets. The CodeXGlue dataset has been used in many recent works; therefore, raising questions about the dataset's quality needs more careful evaluation.
> > - "the idea can be applied to big transformer-based language models easily." - instead of saying, if authors could show the value, it would be helpful.
> >
> > A couple of important points:
> >
> > - Figure 1 shows the proposed model can tell richer information, even more information than the ground truth - but how would BLEU metric reflect that? To support this claim, authors are certifying that "most of the references in the dataset are informative enough". How did they verify that?
> > - Ablation study confirms the value of specific components in the model, I agree. But the author does not verify if the proposed model learns "pyramid-shaped input representation". For example, from the vector representations of the source code tokens, can we decode back the pyramid shape of the input code? There is no such evidence.

---

> > > ### Author Response · Authors · 2022-08-09
> > > **Clarify the disagreements/questions**
> > >
> > > Thanks for your response. We try to clarify the disagreements/questions below:
> > >
> > > **Dataset usage**: There are many source code summarization datasets, none of which are mandatory for this topic. Following recent works in this area, we simply select the relatively accessible, high-quality, and most commonly used datasets for model evaluation.  And we do agree that investigating the generation of PA-former in various languages is an interesting direction.
> > >
> > > What's more, There are many works only evaluated in one dataset[1,2] or one language [3]. All of them are accepted by the top conferences in the field.
> > >
> > > For the sake of judging dataset quality, we simply sample some examples to judge the naturaness of the comments and the consistency between code and comments manually. As for more objective and comprehensive evaluations of the source code summarizaiton datasets, it's another topic beyond the scope of this article.
> > >
> > > **Results on big models**: We don't overclaim anything in terms of big models. and we've already explained that it's easy to apply to transformer-based models and shown that a deeper model can achieve higher performance in Appendix B Table 9. As for more experiment results on big models, this is beyond the scope of this article again.
> > >
> > > **Example in Figure 1**: We just use the case in Figure 1 as an example (it's simple and easy-to-understand) to demonstrate that our model can tell richer information for the given code thanks to the multi-granularity formulation. While the relative "bad" reference (human-written summary) doesn't matter here.
> > >
> > > By the way, we think the reason why our model can predict this informative comment is exactly there are enough informative examples in the dataset and our model effectively capture this pattern.
> > >
> > > What's more, we have done human evaluation and the results also suggest the effectiveness of the method.
> > >
> > > **Pyramid-shaped input representation**:
> > >
> > > The key points behind the proposed pyramid-shaped representation are:
> > >
> > > - the tokens(or elements) at different granularity, which help the model to capture patterns from different scales.
> > > - the relations(or dependencies) between tokens at different scales. which introduces the inductive bias to help the model learn more accurate semantic features
> > >
> > > We design our model based on this cognition. The *bottom-to-top aggregation* cares relations and the *cross-granularity interaction* concerns the tokens at different granularity. And ablation study demonstrates the effectiveness of our design.
> > >
> > > Finally, We are confused about the topic of "decode back the pyramid shape of the input code" why and how?
> > >
> > > **reference**
> > >
> > > 1. A. LeClair, S. Haque, L. Wu, and C. McMillan. Improved code summarization via a graph neural network. In Proceedings of International Conference on Program Comprehension (ICPC), 2020.
> > > 2. Y. Liang and K. Q. Zhu. Automatic generation of text descriptive comments for code blocks. In Proceedings of the 32nd AAAI Conference on Artiﬁcial Intelligence (AAAI), 2018.
> > > 3. E. Shi, Y. Wang, L. Du, H. Zhang, S. Han, D. Zhang, and H. Sun. Cast: Enhancing code summarization with hierarchical splitting and reconstruction of abstract syntax trees. In Proceedings of the 2021 Conference on Empirical Methods in Natural Language Processing (EMNLP), 2021.

---

> > > > ### Comment · Reviewer_Q2Zz · 2022-08-09
> > > > **Decode back the pyramid shape of the input code**
> > > >
> > > > The pyramid attention encourages vector representations to encode the pyramid structure of the code tokens. This could be validated by probing the token representation and evaluating the accuracy of identifying the hierarchical relationship among tokens. For example, in NLP literature, studies claim pretrained encoders capture syntax information of sentences. To verify this, [1] showed that we could train probes on top of a fixed encoder to check if they capture the dependency structure of sentences. Similarly, we could check if the encoded representations produced by the model capture the pyramid (hierarchical) structure of code tokens. It would be an exciting thing to examine.
> > > >
> > > >
> > > > [1] A Structural Probe for Finding Syntax in Word Representations. John Hewitt, Christopher D. Manning.

---

> > > > > ### Author Response · Authors · 2022-08-09
> > > > > **Dependency probe**
> > > > >
> > > > > Thanks for the reply!
> > > > >
> > > > > We do agree that _dependency probe_ could be a viable way to test whether the model has learned structural features. However, the _relations(or dependencies) between tokens_ have been told to the model as known information (e.g. $\mathcal{M}_{\mathcal{L} \rightarrow \mathcal{G}}$) in our PA-former and are used to _guide_ the process of information aggregation. We don't think the _dependency probe_  is suitable here because of the problem of _structure information leakage_. The results on the dataset and ablation study are sufficient to demonstrate the effectiveness of our method in the scope of this paper.
> > > > >
> > > > > The learning of structure in text/sequence is still a challenging problem, and we think our PA-former is a good attempt here.
> > > > >
> > > > > Furthermore and interesting, we use an auxiliary task similar to _dependency probe_ to learn the structure of code in the early stage of this project. Here we show the results:
> > > > >
> > > > > |  Bleu  | Rouge-L | Meteor | Precision | recall  |   f1    |
> > > > > |:------:|:-------:|:------:|:---------:|:-------:|:-------:|
> > > > > | 37.788 |  55.195 | 25.280 |   61.675  |  55.755 |  57.014 |
> > > > >
> > > > > The promising improvement encourages us to try the way of telling the _token type and relation_ directly, and then our PA-former is proposed which achieves higher and stable performance.

---

> > > > > ### Author Response · Authors · 2022-08-09
> > > > > **Other problems except for dependency probe**
> > > > >
> > > > > We would like to know if the supplementary reply (partially) addresses your other questions?

---

### Meta-Review · Area_Chair_GcFG · 2022-08-26

**Recommendation:** Accept
**Confidence:** Less certain

**Metareview:**

The paper presents a multi-granularity input represenaion and a pyramid attention mechanism for code summarizaiton tasks. After extensive discussion, the reviewers still cannot agree on accepting or rejecting this paper. The key discussion points and my opinion are summarized in below.


1. Performance improvement -- a few reviewers point out the performance improvement is relatively small (about 1%) compared with baseline. With the additional error bar provided by the authors, it seems to me the improvement is statistically significant. The authors also provide sufficient ablation study to justify the improvement. Although it's arguable if the proposed appracoh is substantial, the progress of AI is often driven by incremental improvement in terms of performance. Therefore, I'm less concerned by this issue.

2. Comparison only on 1 language and 2 datasets. I partially agree with the authors and reviwers Xtyo that the paper already conducted extensive experiments and the merits of the proposed approach are justified. However, I disagree with the attritue that the comparison on 1 language is sufficient given the recent progress of code summariziton. As the proposed approach is mainly justified by empirical comparison, conducting results on a limited dataset raises the concern whether the proposed approach is generalizble to other languages and datasets. It also makes future work harder to compare with this work. I especially disagree the point that some earlier papers only compared on limited datasets. These papers are published earlier than the benchmark CodeXGlue has been released. As most recent baselines are compared on CodeXGlue, there is a need to justify the missing of results on this dataset. Besides, the argument of the dataet is noisy the performance is low do not seem rigorous and reasonable to me.

3. Human evaluation. The authors provide a preliminary study of human evaluation on the generated outputs. On one hand, it shows the proposed approach indeed improve the quality of the summary, but on the other hand, the study requires more rigorous design. I would suggest including the human evaluation on the main text rather than in appendix.

4. Presentation. The paper is mostly well-written and provide nice intuition behind the propose method. However, I also agree with Q2Zz that some statements might overclaim and require justification. The later part of the paper has significant number of typos and require a careful proofread. It's pity that the authors do not take the opportunity during the rebuttal period to revise the paper.

Overall, I think the paper has sufficient merits but still have room to improve.

**Award:**

No

---

### Decision · Program_Chairs · 2022-09-14

Accept